# Is Early Initiation of Maternal Lactation a Significant Determinant for Continuing Exclusive Breastfeeding up to 6 Months?

**DOI:** 10.3390/ijerph20043184

**Published:** 2023-02-11

**Authors:** Desirée Mena-Tudela, Francisco Javier Soriano-Vidal, Rafael Vila-Candel, José Antonio Quesada, Cristina Martínez-Porcar, Jose M. Martin-Moreno

**Affiliations:** 1Department of Nursing, Universitat Jaume I, 12071 Castellón de la Plana, Spain; 2Department of Nursing, Universitat de València, 46007 Valencia, Spain; 3Department of Obstetrics and Gynaecology, Xàtiva-Oninyent Health Department, 46800 Xàtiva, Spain; 4Department of Obstetrics and Gynaecology, Hospital Universitario de la Ribera, 46600 Alzira, Spain; 5Department of Clinical Medicine, Universidad Miguel Hernández, 03202 Elche, Spain; 6Network for Research on Chronicity, Primary Care, and Health Promotion (RICAPPS), 03550 Alicante, Spain; 7Department of Preventive Medicine and Public Health, Universitat de València, 46010 Valencia, Spain; 8Biomedical Research Institute INCLIVA, Clinic University Hospital, 46010 Valencia, Spain

**Keywords:** exclusive breastfeeding, skin-to-skin contact, vaginal birth, caesarean section, early initiation of breastfeeding

## Abstract

Background: The World Health Organization (WHO) recommends early initiation of breastfeeding (EIBF) within the first hour after birth. However, certain perinatal factors, namely caesarean section, may prevent this goal from being achieved. The aim of our study was to examine the relationship between EIBF (maternal lactation in the first hours and degree of latching before hospital discharge) and the maintenance of exclusive breastfeeding (MBF) up to the recommended 6 months of age (as advocated by the WHO). Methods: This observational, retrospective cohort study included a random sample of all births between 2018 and 2019, characterising the moment of breastfeeding initiation after birth and the infant’s level of breast latch (measured by LATCH assessment tool) prior to hospital discharge. Data were collected from electronic medical records and from follow-up health checks of infants up to 6 months postpartum. Results: We included 342 women and their newborns. EIBF occurred most often after vaginal (*p* < 0.001) and spontaneous births with spontaneous amniorrhexis (*p* = 0.002). LATCH score <9 points was associated with a 1.4-fold relative risk of abandoning MBF (95%CI: 1.2–1.7) compared with a score of 9–10 points. Conclusions: Although we were unable to find a significant association between EIBF in the first 2 h after birth and MBF at 6 months postpartum, low LATCH scores prior to discharge were associated with low MBF, indicating the importance of reinforcing the education and preparation efforts of mothers in the first days after delivery, prior to the establishment of an infant feeding routine upon returning home.

## 1. Introduction

The World Health Organization (WHO), in addition to other international organisations, recommends early initiation of breastfeeding (EIBF) during the first hour of life, maintaining exclusive breastfeeding (MBF) until 6 months postpartum, and continuing breastfeeding (BF) after this period in addition to suitable healthy foods for infants until the age of 2 years or older [1,2]. This form of feeding is considered effective in ensuring newborn infants’ health and survival worldwide [2].

To achieve EIBF within the newborn’s first hour of life and to establish adequate BF, immediate skin-to-skin contact (SSC) between the mother and newborn has been identified as an effective healthcare intervention [3,4,5].

Although immediate SSC is widely accepted, there are socio-economic and perinatal factors, especially caesarean section, which make it difficult to achieve this goal, despite knowledge that early SSC is possible after caesarean section, and performing it as early as possible is recommended [6,7]. Different authors have reported the many benefits of SSC and have observed that women wish to perform this practice [8,9]; however, it is unfortunately not widespread. Moreover, a threefold increased risk has been reported for delaying BF initiation in infants born by caesarean section compared with infants born by vaginal birth [10,11]. In addition to birth type, different perinatal and socio-economic factors related to delayed EIBF have been identified, such as intrapartum complications, labour duration, gestational age at birth [11,12], use of anaesthetics and opiates [13], newborn’s sex, birth weight, APGAR score, and being admitted to a neonatal intensive care unit (NICU) [5,14].

Different studies have stressed the importance of analysing the relationship between the perinatal factors that might influence EIFB and the infant’s level of breast latch; however, few studies have observed its relationship with mid-term (<6 months) MBF. For this reason, it remains necessary to verify the relationship between EIBF, effective latching, and MBF until 6 months postpartum. The secondary objectives of this study are to determine the factors influencing EIBF, effective latching, and MBF until 6 months postpartum.

## 2. Materials and Methods

### 2.1. Study Design

An observational, retrospective cohort study was performed. This study was conducted at the University La Ribera Hospital (HULR) in Valencia (Spain), which serves a population of approximately 250,000 inhabitants and sees a mean 1400 births/year.

The inclusion criteria were women aged 18 years or older and all natural births and caesarean sections that took place at the HULR between 2018 and 2019 in which EIBF occurred inside a delivery room after ‘very early SSC’. In line with Moore et al.’s [3] definition, very early SSC was defined as contact beginning approximately 30–40 min postpartum, wherein the naked infant, with or without a cap, is placed prone on the mother’s bare chest and a blanket is placed across the infant’s back.

Twin and multiple births, preterm infants, newborns admitted to a NICU, infants with no record of EIBF within the first 120 min after birth inside a delivery room, and infants with BF-type records lost during the 6-month postpartum follow-up were excluded.

### 2.2. Data Collection

Data collection was based on the electronic medical records. All children born in the hospital with infant feeding records were included. The study sample was selected by simple random sampling. The response variable was infant feeding. Feeding type upon hospital discharge was classified and grouped as BF, artificial, or mixed. Infants who abandoned exclusive BF were assigned to the artificial or mixed group. Milk feeding was measured at 1, 2, 4, and 6 months postpartum. Exclusive MBF was defined as infants who were fed exclusively drawn/donor breast milk from the mother. In addition, MBF newborns only received vitamin drops or syrups, medications, or minerals [15]. Artificial feeding was defined when the breastfed infant was fed only with artificial milk, and mixed feeding was defined when an infant’s feeding combined BF and artificial milk.

The study variables were: (1) socio-demographic (maternal age and country of origin) and (2) obstetric (gestational age, parity, pregnancy risk (based on the Spanish Society of Gynaecology and Obstetrics, classification as low, medium, high/very high determined by healthcare provider. Different factors, such as maternal age, previous medical conditions, previous or actual obstetric history, and lifestyle factors, may affect the mother´s health and/or the developing foetus [15]), birth initiation, amniorrhexis type, analgesia, and end of birth); following the recommendations of Devane et al. [16], (3) perinatal variables (newborn’s sex, birth weight, birth length, cephalic perimeter, umbilical artery pH), and (4) feeding (LATCH breastfeeding assessment tool [17], EIBF time and feeding type) were included. The time until EIBF was recorded by the midwife who assisted the birth as routine data in the electronic medical record and was categorised into two periods, ≤60 min or >60 min, with a maximum time of 120 min. The LATCH score was measured and recorded on the date of discharge from the hospital. This LATCH scale measures BF efficiency using five items. Each item is given a maximum score of 2 points and a minimum score of 0, with a maximum 10-point score (the acronym LATCH corresponds to: L ‘how well infant latches onto the breast’, A ‘audible swallowing’, T ‘type of nipple’, C ‘comfort’, and H ‘hold-positioning’). To optimise the LATCH analysis, LATCH scores were categorised into two categories: <9 points and 9–10 points. As reported by other authors, a LATCH score ≥8 at 48 h or discharge had a sensitivity of 93.5% and specificity of 92.1%, with these mothers being 9.28 times more likely to BF at 6 weeks postpartum [17].

Sample size was calculated by assuming a 50% MBF prevalence at 6 months postpartum with 5% precision, a 95% confidence interval (95%CI), and an expected 10% proportion of losses. The sample required 335 women. Finally, randomisation of medical record numbers was performed for the births that occurred during the study period, assigning every individual a number by using a random number generator and then randomly picking a subset of the estimated population.

### 2.3. Data Analysis

Descriptive analysis was performed with all the variables by calculating frequencies for the qualitative variables, minimums, maximums, and mean ± standard deviation (SD) for the quantitative variables.

To analyse how abandoning BF evolved between 1 and 6 months, a simple linear regression line was adjusted to the proportion of abandonments per month. The *R*^2^ association was calculated. The factors associated with the presence of an at-risk LATCH (<9 points) [17], EIBF after >1 h, and abandoning MBF at 6 months postpartum were analysed using contingency tables and by applying the Chi-squared test for the qualitative variables. The mean values for the quantitative variables were compared using Student’s *t*-test.

To estimate the magnitude of the associations with abandoning MBF at 6 months, relative risks (RRs) were estimated using Poisson regression models with robust variance [18] in addition to their 95% CIs. A stepwise procedure was followed to select the variables based on the Akaike Information Criterion. Data analysis was performed using SPSS v.26.0 for Windows (IBM Corp. 2018, Armonk, NY, USA) and R (R project 2019, version 4.0.2). As the analysis included two variables (EIBF and abandoning MBF), the level of significance was adjusted by the Bonferroni method to *p* < 0.025.

### 2.4. Ethical Considerations

This study received a favourable opinion from the Research Ethics Committee and the Research Committee (CEI-CI) of HULR (HULR2020_34). The bioethical principles of the Declaration of Helsinki were applied. 

## 3. Results

Of the 1104 cases with early SSC, an EIBF record, and mothers wishing to undertake postpartum BF, 497 women were randomly included. Due to loss through follow-up at the different cut-off points, 155 cases were omitted (Figure 1). We were interested in analysing the missing cases to determine whether there were statistically significant differences between them, especially regarding the type of BF at discharge. There were no significant differences between the group of women lost to follow-up and type of BF at discharge regarding age, gestational age at delivery, newborn´s sex, pregnancy risk, country of origin, analgesia, or birth started. In contrast, there were significant reductions among the participants with lower LATCH scores (*p <* 0.01), type of amniorrhexis (*p <* 0.01), and end of birth (*p <* 0.01). In addition, there was a significant difference between this group and the type of BF at discharge, where women who had EIBF (≤60 min) with exclusive MBF at discharge (EIBF ≤60 min with MBF [*n* = 106], with abandonment of exclusive BF [*n* = 15]; EIBF >60 min with MBF [*n* = 20] and with abandonment of exclusive BF [*n* = 14]; *p* < 0.001).

The final analysed sample contained 342 women (Table 1) with a mean age of 32.9 ± 5.4 years, of whom 87.7% (300/342) were Spanish. The mean gestational age at birth was 39+4 ± 1.2 weeks, and 38.9% (133/342) had a caesarean section, 55.8% (191/342) were primiparous, and 29.5% (101/342) had a high-risk or very high-risk pregnancy.

Of all the newborns, 51.2% (175/342) were boys, and the mean birth weight was 3351 ± 485 g. The recorded valid EIBF rate was 11.9% (342/2879), with 87.7% (300/342) of mothers performing EIBF during the first hour of life. Moreover, 20.2% (69/342) obtained a high-risk LATCH score (<9 points). Abandoning MBF increased with newborn’s age to 26.0% (89/253), 34.5% (118/224), 43.9% (150/192), and 61.4% (210/132) at 1, 2, 4, and 6 months postpartum, respectively. A linear trend was observed for abandoning MBF up to 6 months postpartum, with a 6.79% mean proportion for abandonment per month (linear *R*^2^ = 0.984). 

Table 2 shows the results of analysing the profiles of mothers with EIBF according to EIBF time (≤60 min or >60 min). EIBF within 1 h of life occurred most often with vaginal (*p* < 0.001) births and spontaneous births with spontaneous amniorrhexis (*p* = 0.002). Moreover, there was an association between EIBF and LATCH score (*p* = 0.023), where 89.7% (245/342) of newborns with scores of 9–10 received EIBF before 60 min. The prevalence of MBF at 1, 2, 4, and 6 months was 74.0% (253/342), 65.5% (224/342), 56.1% (192/342), and 38.6% (132/342), respectively. No significant differences were found between MBF at 1, 2, 4, and 6 months and the quantitative variables.

Table 3 shows how significantly more mothers who abandoned MBF at 6 months postpartum obtained LATCH scores <9 points (81.2%) than scores of 9–10 points (56.4%) (*p* < 0.001). No differences were observed between birth types regarding abandoning MBF at 6 months postpartum; 61.3% (82/133) of mothers abandoned MBF after vaginal birth, and 61.6% (128/209) abandoned MBF after caesarean section (*p* = 0.939). No significant differences were found for any quantitative variable.

Table 4 shows the multivariate analysis used to calculate the RR of abandoning MBF at 6 months postpartum. We found no optimum model to explain MBF abandonment at 6 months postpartum because no factors were associated with this phenomenon. Only a LATCH score <9 points was associated with a 1.4-fold (1.2–1.7) increased RR of abandoning MBF compared with a score between 9–10 points adjusted for EIBF time. EIBF did not function as either a confounder or a modifying variable in this association; therefore, the LATCH score was independently related to the maintenance or abandonment of exclusive MBF.

## 4. Discussion

EIBF favours MBF after hospital discharge; however, we did not find a significant association between EIBF and MBF at 6 months postpartum. The lack of a significant relationship does not imply that the relationship does not exist. It is possible that the abandonment was caused by social or environmental variables and/or support policies aimed at maintaining BF that affected the maintenance of BF in the long term to a greater extent [13,19]. However, we clearly and significantly observed a correlation between the LATCH score assessed before hospital discharge and the maintenance of MBF at 6 months, and this relationship was statistically independent from whether or not EIBF had occurred in the first hour of the newborn’s life. This finding suggests that the LATCH assessment should not be considered simply a collection of information, but rather a proactive element in the education and enabling intervention for the mother before returning home. If the mother is able to manage satisfactory BF autonomously, it is more likely that exclusive BF will continue until the sixth month, thereby achieving the goal that is clearly defined as ideal by the scientific literature and endorsed by the WHO and UNICEF [20].

Regarding the relationship between EIBF and birth type, we observed how EIBF took longer to initiate in newborns after caesarean section than in those born by vaginal birth [4]. This finding corroborates other studies that observed that women with immediate SSC were more likely to initiate early BF in different modes of birth [11,21,22,23,24]. After caesarean section at our hospital, early SSC is performed by the women chosen by the mother until she arrives. Although SSC is not usually performed later than 15 min after birth, it could be started earlier if early SSC was implemented in caesarean section-type births. A delay in initiating BF could be explained by characteristics related to surgery and could be due to the mothers’ initial post-surgical recovery. A negative birth experience, fatigue, and anxiety, especially in the context of an urgent caesarean section, may result in mothers needing to rest, which can delay the initiation of BF [11]. End of birth type (vaginal vs. caesarean section) did not have a significant effect on abandoning MBF early at 6 months, as reported in other publications [4,11]. Regarding the time to initiate BF, a shorter time between birth and BF initiation was not associated with increased long-term BF rates, which is in line with the results of other studies [5]. BF is an instinctive act of the healthy newborn. The time pressure for the infant’s first latch-on should be relativised. In this regard, it is important to highlight the importance of early SSC between the mother and newborn regardless of the type of childbirth or the gestational age and weight of the baby, as recommended by the WHO [25]. Health professionals should focus on allowing time and space for this dyad and observing and being available for maternal concerns and hand off.

Regarding the relationship between EIBF and number of children, the likelihood of EIBF increased as parity increased. The mother’s experience and empowerment may play a key role [26], where empowerment is a tool that motivates women with autonomy to employ resources and to overcome structural or social limitations [26]. In addition, health literacy might play a role [27]. Nonetheless, future studies should confirm these hypotheses.

The LATCH assessment tool is not designed to make long-term predictions; however, other studies have indicated its possible predictive value as a variable associated with the continuation of long-term MBF [17,28]. In addition, our study observed how LATCH scores below 9 points favoured MBF abandonment at 6 months, which was also associated with lower birth weight and cephalic perimeter values. This observation opens an interesting line of future research in which the LATCH assessment tool could function as a predictor of long-term BF.

The multivariate model indicated that the LATCH tool intervened in the continuation of long-term MBF by demonstrating that EIBF did not affect this association. Healthcare professionals’ continuous postpartum support forms a fundamental part of EIBF [5,12,13]. Perhaps, our efforts must concentrate on constantly providing mothers with professional support as needed for them to discover a comfortable and painless BF position after either vaginal birth or caesarean section (e.g., placing the newborn in a cross-sectional position if born by caesarean section to optimise SSC and to improve spontaneous latching). This could indirectly improve the LATCH score and help mothers to continue with long-term MBF [11]. Because factors, such as artificial feeding and birth by caesarean section, are associated with the development of long-term diseases, such as multiple sclerosis [29], efforts made to improve BF as a future health intervention are the minimum, especially as it involves inexpensive healthcare measures with excellent and positive outcomes for the health of mothers and their babies. Nonetheless, this line of inquiry would benefit from future research.

In view of our results and in line with other studies [30], it is necessary to abandon the idea that starting BF after caesarean section is more difficult. Working with a multidisciplinary team to raise the awareness of healthcare professionals who attend women giving birth by caesarean section would be a relevant area to investigate. In addition, more respectful healthcare in line with Baby-Friendly Hospital Initiative (BFHI) recommendations would mark the difference with the related variables. It is essential to underline that accreditation by the BFHI does not ensure efficient support [31]; however, it is a step in this direction.

Despite the findings presented in this study, it had several limitations. Therefore, our results must be carefully interpreted, and we must consider the impact of follow-up losses on the true prevalence of MBF. More cases were lost than the initially estimated sample because our study was retrospective, and some electronic medical records were missing. Although this limitation is common in retrospective studies, our study was able to reach the estimated sample size. Moreover, the study sample’s randomisation avoided selection bias, which improved the comparison between EIBF and end of birth type (vaginal vs. caesarean section).

Furthermore, we would like to emphasise that determining the causes of abandonment of MBF would have produced interesting data, although this could only have been investigated if the study had a prospective design; unfortunately, it was not part of our objective to analyse the causes of MBF abandonment. It should be noted that the results could have been improved if the SSC guidelines had been followed, achieving immediate SSC (within 10 min) and avoiding the use of a cap on the newborn [3].

## 5. Conclusions

Although EIBF as soon as possible is considered a priority, in this study, it was not associated with MBF at 6 months postpartum. However, we found that low LATCH scores were associated with cessation at 6 months. Any health intervention capable of improving LATCH scores would imply higher rates of the desired MBF target, and we have the professional and ethical responsibility to reinforce education and empowerment interventions for the mother before she returns home and establishes her infant feeding routine to facilitate the path towards achieving the goal of MBF until the sixth month, which has proven to be beneficial for both the child and the mother.

## Figures and Tables

**Figure 1 ijerph-20-03184-f001:**
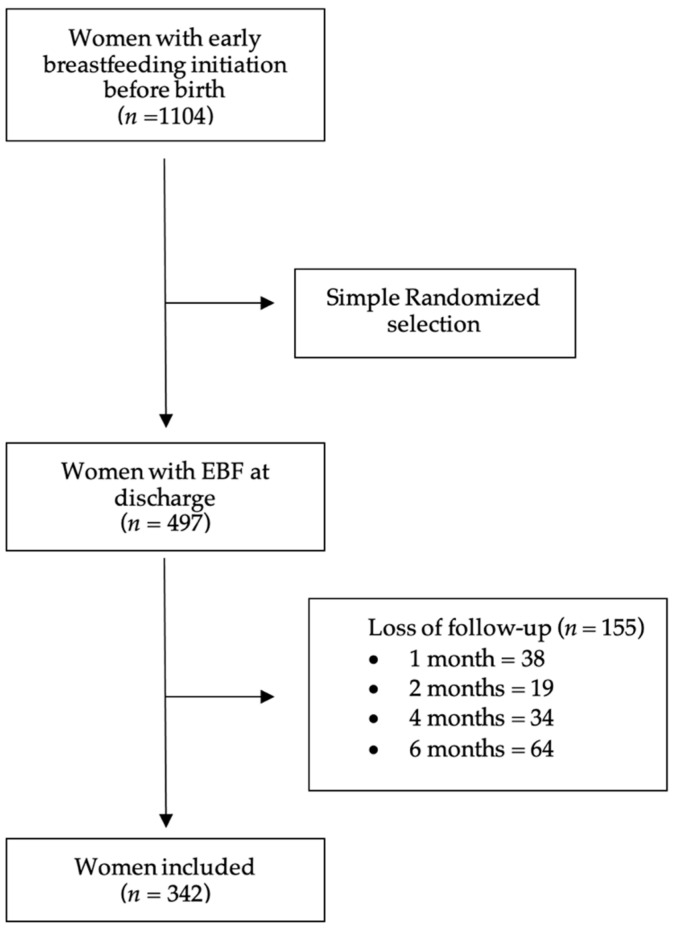
Flow chart of the study. MBF: exclusive breastfeeding.

**Table 1 ijerph-20-03184-t001:** The studied sample’s characteristics (*n* = 342).

					*n* %
Abandoning MBF ^a^ 1 Month		No		253	74.0%
		Yes		89	26.0%
Abandoning MBF ^a^ 2 months		No		224	65.5%
		Yes		118	34.5%
Abandoning MBF ^a^ 4 months		No		192	56.1%
		Yes		150	43.9%
Abandoning MBF ^a^ 6 months		No		132	38.6%
		Yes		210	61.4%
LATCH ^b^		9–10 Normal		273	79.8%
		< 9 Risk		69	20.2%
EIBF ^c^		≤ 60 min		300	87.7%
		> 60 min		42	12.3%
Newborn’s sex		Male		175	51.2%
		Female		167	48.8%
Country of origin		Others		42	12.3%
		Spain		300	87.7%
Parity		One		191	55.8%
		Two		119	34.8%
		Three or more		32	9.4%
Pregnancy risk		Low		199	58.2%
		Medium		42	12.3%
		High/Very high		101	29.5%
Analgesia		Epidural		240	70.2%
		Others		102	29.8%
Birth started by		Stimulated		42	12.3%
		Induced		77	22.5%
		Caesarean		42	12.3%
		Spontaneous		181	52.9%
End of birth		Vaginal		209	61.1%
		Caesarean		133	38.9%
Amniorrhexis		Artificial		211	61.7%
		Spontaneous		131	38.3%
	** *n* **	**Minimum**	**Maximum**	**Mean**	**SD ^d^**
Maternal age	342	18.0	51.0	32.9	5.4
Gestational age (weeks)	342	37.0	42.0	39.4	1.2
Birth weight (g)	342	2195.0	5730.0	3351.5	485.0
Birth length (cm)	342	43.0	55.5	50.0	1.9
Cephalic perimeter (cm)	342	31.0	48.0	34.5	1.7
Umbilical artery pH	306	7.08	7.43	7.3	0.1

^a^ MBF: exclusive breastfeeding; ^b^ LATCH: breastfeeding assessment tool score; ^c^ EIBF: early initiation of breastfeeding; ^d^ SD: standard deviation.

**Table 2 ijerph-20-03184-t002:** Relation between early latching and the other variables (*n* = 342).

		EIBF ^a^ ≤ 60 min		EIBF ^a^ > 60 min		
		*n*	%	*n*	%	*p*-Value ^b^
Newborn’s sex	Male	158	90.3%	17	9.7%	0.139
	Female	142	85.0%	25	15.0%	
Country of origin	Others	41	97.6%	1	2.4%	0.037
	Spain	259	86.3%	41	13.7%	
Parity	One	160	83.8%	31	16.2%	0.043
	Two	110	92.4%	9	7.6%	
	Three or more	30	93.8%	2	6.3%	
Pregnancy risk	Low	174	87.4%	25	12.6%	0.983
	Medium	37	88.1%	5	11.9%	
	High/very high	89	88.1%	12	11.9%	
Analgesia	Epidural	211	87.9%	29	12.1%	0.865
	Others	89	87.3%	13	12.7%	
Birth started by	Caesarean	35	83.3%	7	16.7%	0.155
	Spontaneous	165	91.2%	16	8.8%	
	Stimulated	37	88.1%	5	11.9%	
	Induced	63	81.8%	14	18.2%	
End of birth	Vaginal	197	94.3%	12	5.7%	<0.001
	Caesarean	103	77.4%	30	22.6%	
Type of amniorrhexis	Artificial	176	83.4%	35	16.6%	0.002
	Spontaneous	124	94.7%	7	5.3%	
LATCH	9–10 Normal	245	89.7%	28	10.3%	0.023
	<9 Risk	55	79.7%	14	20.3%	
MBF 1 month	Yes	219	86.6%	34	13.4%	0.271
	No	81	91.0%	8	9.0%	
MBF 2 months	Yes	195	87.1%	29	12.9%	0.605
	No	105	88.9%	13	11.1%	
MBF 4 months	Yes	165	85.9%	27	14.1%	0.256
	No	135	90.0%	15	10.0%	
MBF 6 months	Yes	115	87.1%	17	12.9%	0.789
	No	185	88.1%	25	11.9%	

^a^ EIBF: early initiation of breastfeeding; ^b^ Chi-squared test.

**Table 3 ijerph-20-03184-t003:** Abandoning exclusive breastfeeding at 6 months postpartum (*n* = 342).

		Not Abandoned MBF ^a^ 6 Months		Abandoned MBF ^a^ 6 Months		
		*n*	%	*n*	%	*p*-Value ^b^
LATCH ^c^	9–10 Normal	119	43.6%	154	56.4%	< 0.001
	<9 risk	13	18.8%	56	81.2%	
EIBF ^d^	≤60 min	115	38.3%	185	61.7%	0.789
	>60 min	17	40.5%	25	59.5%	
Newborn’s sex	Male	67	38.3%	108	61.7%	0.904
	Female	65	38.9%	102	61.1%	
Country of origin	Others	15	35.7%	27	64.3%	0.682
	Spain	117	39.0%	183	61.0%	
Parity	One	71	37.2%	120	62.8%	0.580
	Two	46	38.7%	73	61.3%	
	Three or more	15	46.9%	17	53.1%	
Pregnancy risk	Low	76	38.2%	123	61.8%	0.422
	Medium	13	31.0%	29	69.0%	
	High/very high	43	42.6%	58	57.4%	
Analgesia while giving birth	Epidural	94	39.2%	146	60.8%	0.740
	Others	38	37.3%	64	62.7%	
Birth started by	Caesarean	15	35.7%	27	64.3%	0.616
	Spontaneous	72	39.8%	109	60.2%	
	Stimulated	19	45.2%	23	54.8%	
	Induced	26	33.8%	51	66.2%	
End of birth	Vaginal	81	38.8%	128	61.2%	0.939
	Caesarean	51	38.3%	82	61.7%	
Type of amniorrhexis	Artificial	78	37.0%	133	63.0%	0.432
	Spontaneous	54	41.2%	77	58.8%	

^a^ MBF: exclusive breastfeeding; ^b^ Chi-squared test; ^c^ LATCH: breastfeeding assessment tool score; ^d^ EIBF: early initiation of breastfeeding.

**Table 4 ijerph-20-03184-t004:** Multivariate analysis of abandoning exclusive breastfeeding at 6 months postpartum.

		Simple Adjustment			Multivariate Adjustment		
		RR ^a^	95%CI ^b^	*p*-Value ^c^	RR ^a^	95%CI ^b^	*p*-Value ^c^
LATCH ^d^	≥9 Normal	1			1		
	<9 Risk	1.44	(1.23–1.67)	<0.001	1.45	(1.24–1.70)	<0.001
EIBF ^e^	≤60 min	1			1		
	>60 min	0.97	(0.74–1.26)	0.794	0.91	(0.71–1.17)	0.458

^a^ RR: relative risk; ^b^ CI: confidence interval; ^c^ Poisson multiple regression model; ^d^ LATCH: breastfeeding assessment tool; ^e^ EIBF: early initiation of breastfeeding.

## Data Availability

Data are available upon reasonable request. All necessary data are supplied and available in the manuscript; however, the corresponding author will provide the dataset upon request. All data relevant to the study are included in the article.

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
