# Peer review of "Is Early Initiation of Maternal Lactation a Significant Determinant for Continuing Exclusive Breastfeeding up to 6 Months?"

_ijerph, 2023, doi:10.3390/ijerph20043184_

Round 1

Reviewer 1 Report

The manuscript is interesting and the subject important.

I have some questions and suggestions.

1. page 4. There is nmo information about the randomization procedure. Please complete and argue for why you did use the randomization instead of using the total sample of 1041 individuals.

2. page 7, table 2. In the column of EIFF and % there seems to be a mistake for MBF 1, 2, 4 and 6 months were % should be written. It should be 73% instead of 0,73 for MBF 1 mo etc.

Author Response

Response to Reviewer 1 Comments

Thank you for your very positive and constructive feedback on our manuscript.

We have considered all your comments and suggestions (and also the comments made by the other reviewer) attempting to improve/refine the original manuscript.

Below you will find a point-by-point response to your comments (in red).

The manuscript is interesting and the subject important.

I have some questions and suggestions.

Point 1: page 4. There is no information about the randomization procedure. Please complete and argue for why you did use the randomization instead of using the total sample of 1041 individuals.

Response 1: Thank you for your comment. An explanation of the randomization method has been included in the Material and Methods section (lines 118-120). Thank you for giving us the opportunity to explain that simple random sampling is used to make statistical inferences about a population. It helps ensure high internal validity randomization is the best method to reduce the impact of potential confounding variables.

Point 2: page 7, table 2. In the column of EIFF and % there seems to be a mistake for MBF 1, 2, 4 and 6 months were % should be written. It should be 73% instead of 0,73 for MBF 1 mo etc.

Response 2: Thank you for spotting that error. Amended.

Reviewer 2 Report

The study by Mena-Tudela et al. investigates the important correlation the relationship between EIBF, effective latching and breastfeeding until 6 months postpartum. We need these studies to define where the mothers need more support.

The retrospective study was carried out with a good design. The selection of the study population by simple random sampling is very suitable.

The discussion is well structured, although some speculations cannot be supported by the data obtained from the study.

Special comments:

Material and methods:

Page 2, line 86: The reference to Figure 1 should be deleted. Figure 1 shows a result.

Page 3, line 99: please define pregnancy risk. What does that mean?

Results:

Page 4, line 149: please define risk of pregnancy. Is that the same as pregnancy risk?

Risk pregnancy or risk of pregnancy is also used in Tables 1 and 3. Please use one term consistently.

Author Response

Response to Reviewer 2 Comments

Thank you for your positive comments on our manuscript and for your constructive remarks.

We have considered all your comments and suggestions (and also the comments made by the other reviewer) attempting to improve/refine the original manuscript.

Below you will find a point-by-point response to your comments (in red).

The study by Mena-Tudela et al. investigates the important correlation the relationship between EIBF, effective latching and breastfeeding until 6 months postpartum. We need these studies to define where the mothers need more support. 

The retrospective study was carried out with a good design. The selection of the study population by simple random sampling is very suitable.

The discussion is well structured, although some speculations cannot be supported by the data obtained from the study.

Special comments:

Material and methods:

Point 1: Page 2, line 86: The reference to Figure 1 should be deleted. Figure 1 shows a result.

Response 1: Thank you for the observation and suggestion. Accordingly, it has been deleted in line 86.

Point 2: Page 3, line 99: please define pregnancy risk. What does that mean?

Response 2: Thank you for your comment. We have included the three categories and a reference number 15 where the pathologies affecting each of the risks are specified (https://sego.es/documentos/progresos/v61-2018/n5/GAP_Control%20prenatal%20del%20embarazo%20normal_6105.pdf).  We hope this clarifies this aspect.

Results:

Point 3: Page 4, line 149: please define risk of pregnancy. Is that the same as pregnancy risk?

Response 3: Thanks for your suggestion and question. The correct term is pregnancy risk and not risk of pregnancy. Amended on line 151.

Point 4: Risk pregnancy or risk of pregnancy is also used in Tables 1 and 3. Please use one term consistently.

Response 4: Thank you again. The changes have been amended as pregnancy risk.